# SCALING OPEN-WORLD MULTIPLE OBJECT TRACKING

## ABSTRACT

Multiple Object Tracking (MOT) has traditionally relied on expensive, exhaustively annotated datasets, limiting scalability and generalization. To address these limitations, we propose **ScaleTrack**, a transformer-based association module for MOT, explicitly designed to leverage large-scale, sparsely annotated video data. At the core of our approach is *Chain Contrastive Learning*, a novel contrastive strategy that maintains local discriminability while capturing long-range temporal coherence. Specifically, our approach constructs positive pairs in a chained manner across consecutive frames, promoting transitive consistency and local discriminability simultaneously. Our model additionally features a multi-scale spatiotemporal attention mechanism that effectively integrates contextual information across space and time, ensuring robust associations even in challenging scenarios. Notably, our method consistently improves performance as the amount of training video data increases, demonstrating robust scalability. Our tracker is designed as a plug-and-play module that seamlessly synergizes with any object detector, achieving state-of-the-art zero-shot performance across multiple large-scale MOT benchmarks, including TAO, BDD100K, SportsMOT and OVT-B. Code will be made public.

## 1 INTRODUCTION

Multiple Object Tracking (MOT) is crucial for applications like autonomous driving and robotics Sun et al. (2020a); Caesar et al. (2020); Yu et al. (2020); Grauman et al. (2022), but modern systems fail to generalize to new, unseen environments. The key to robust, open-world performance is scaling on massive, diverse video datasets. Exhaustively annotating every object in large, diverse video collections is prohibitively difficult and expensive. Consequently, datasets like SA-V, TAO Ravi et al. (2024); Dave et al. (2020) are only sparsely annotated, which presents a core challenge: finding a training recipe that can effectively leverage this data and, crucially, scale well with it.

Current MOT paradigms are difficult to scale. End-to-end transformers Zeng et al. (2022); Meinhardt et al. (2022); Zhang et al. (2023b); Luo et al. (2023); Gao & Wang (2023), with their joint detection and tracking mechanisms, require exhaustive annotations for every object, every class. This makes them incompatible with the non-exhaustive labels of large-scale video datasets, limiting their scaling on rich resources. The classic tracking-by-detection paradigm also falls short; its simple motion models are often not robust enough for complex, real-world dynamics. This leaves a powerful, learnable appearance association module as the most viable path toward generalizable MOT.

However, the dominant contrastive learning (CL) strategies used to train such models also fail the crucial test of scalability (Fig. 1a). Firstly, pairwise or local methods Pang et al. (2021); Wu et al. (2022); Li et al. (2022); Yan et al. (2023); Wu et al. (2023), contrast an object only with counterparts in adjacent frames. While effective for short-term discriminability, this "myopic" approach yields features less robust to prolonged occlusions or substantial appearance variations over time, and because each positive pair only covers a short temporal window, the amount of useful learning signal grows much more slowly than the data volume, causing rapid saturation as data scale up. Secondly, *global* contrastive strategies De Plaen et al. (2024) aim for long-term coherence by comparing an instance against diverse negatives from extensive temporal contexts (akin to principles in, *e.g.*, SupCon Khosla et al. (2020) or InfoNCE Oord et al. (2018) from other domains). However, when directly applied to tracking, enforcing discrimination based on isolated snapshots from vastly different contexts can be problematic. True instance identity can be ambiguous without continuous observation (*e.g.*, distinguishing identical objects is trivial if in different rooms, but requires subtle local cues if they are adjacent). Such global comparisons risk misdirecting feature learning towards coarse,

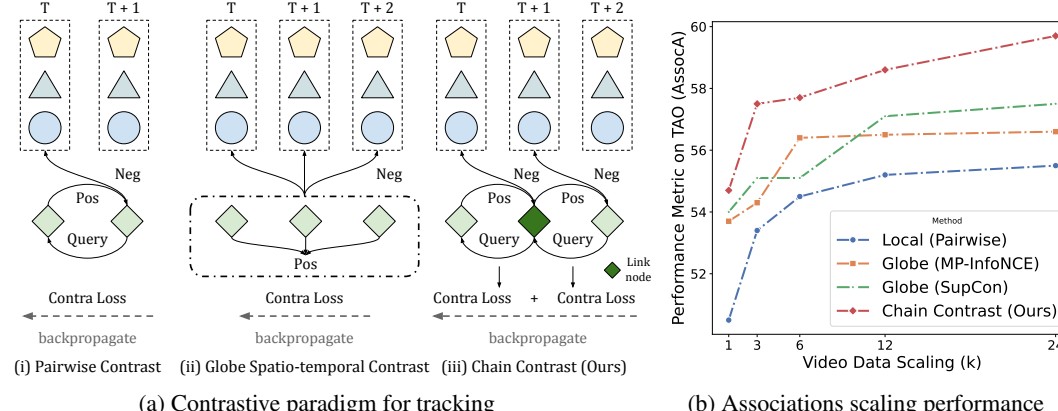

(a) Contrastive paradigm for tracking          (b) Associations scaling performance

Figure 1: (a) **Contrastive paradigms for tracking.** (i) *Pairwise* uses only adjacent frames, limiting long-term learning. (ii) *Global* contrasts all instances across a batch, promoting long-range coherence but introducing many off-context negatives that hurt local discriminability. (iii) *Chain Contrast* links instances through consecutive frames, balancing local precision with long-range consistency. (b) **Scaling on TAO.** Chain Contrast (red) attains higher AssocA and scales better with more video than Pairwise or Global.

scene-level contextual differences rather than honing the precise, local discriminability essential for frame-to-frame association, or may introduce less relevant negative pairs that hinder learning. As the data size increases, the number of irrelevant or ambiguous negatives grows rapidly, diluting the training signal and causing the learning to plateau. As we demonstrate in Fig. 1b, both local and global approaches plateau, failing to translate more data into better performance.

To break this scaling barrier, we introduce **Chain Contrastive Learning (CCL)**, a temporal learning framework designed explicitly to thrive on large-scale data. CCL builds long-range temporal consistency by transitively linking an object's identity across a sequence of multiple frames. Crucially, negatives are drawn only from the local spatio-temporal context, forcing the model to resolve fine-grained ambiguities against the most relevant distractors. This unique design avoids the saturation of local methods and the signal dilution of global ones, enabling performance to grow with data.

With the improved representational power of CCL, we present **ScaleTrack**, a dedicated transformer-based tracking framework designed to effectively extract and utilize these powerful appearance features. ScaleTrack leverages a frozen pretrained backbone (*e.g.* DINOv2 Oquab et al. (2024)) for rich initial features, adapted by a transformer encoder. Unlike prior MOT architectures Zeng et al. (2022); Meinhardt et al. (2022) that entangle detection and tracking queries, our framework uses queries *purely* for extracting dedicated appearance embeddings for association. These embeddings are refined through spatial attention pooling and a hybrid decoder that cross-attends to multi-scale spatial features from the current frame and a temporal buffer of past contexts, enabling robust association. To further maximize learning from diverse videos with non-exhaustive annotations, CCL within ScaleTrack is augmented by incorporating pseudo-negative proposals from foundation models (*e.g.* Detic Zhou et al. (2022)), enriching the set of hard local negatives.

Our main contributions are: (1) Chain Contrastive Learning, a new temporal contrastive framework that robustly models long-range appearance evolution while preserving local discriminability for MOT, significantly outperforms existing contrastive paradigms in both association accuracy and scalability with increased video data. (2) The ScaleTrack tracking architecture, designed to effectively implement CCL by extracting and fusing rich spatio-temporal appearance cues for association.

Our approach achieves strong zero-shot generalization on challenging open-vocabulary MOT benchmarks Li et al. (2023), even outperforming in-domain trained baselines on specialized datasets like BDD100K Yu et al. (2020) and SportsMOT Cui et al. (2023).

## 2 RELATED WORK

**Learning instance-level associations.** is crucial to multiple object tracking (MOT), with existing methods primarily falling into supervised and self-supervised categories Pang et al. (2021); Wu et al. (2022); Yan et al. (2023); Li et al. (2022); Zhang et al. (2023b); Zeng et al. (2022); Meinhardt et al. (2022); Wang et al. (2020); Yan et al. (2022). Self-supervised approaches, such as UniTrack Wang et al. (2021b) and MASA Li et al. (2024a), leverage generic representations learned from unlabeled images Chen et al. (2020); Xu & Wang (2021). Other self-supervised methods, like TCC Dwibedi et al. (2019), learn from unlabeled videos by enforcing a forward-and-backward cycle, though these are typically aimed at correspondence rather than the fine-grained identity discrimination required in MOT. While providing competitive results, these methods can be limited in handling challenging temporal dynamics. Alternative methods attempt to reduce annotation dependency via synthetic data Li et al. (2023), training from static images Zhou et al. (2020); Fu et al. (2021); Athar et al. (2022), or test-time adaptation Segu et al. (2023), yet these strategies remain insufficient for modeling realistic temporal variations in diverse scenarios. Supervised methods predominantly employ contrastive learning on annotated image pairs sampled from videos Pang et al. (2021); Wu et al. (2022); Yan et al. (2023); Li et al. (2022); Fang et al. (2024), achieving superior performance in specific domains. Such *local contrastive* methods effectively capture short-term discriminability but struggle to scale effectively, as the long-range temporal context in videos is not exploited. Alternative approaches attempt to utilize global supervised contrastive losses Khosla et al. (2020); Oord et al. (2018); Fang et al. (2024), contrasting objects across entire video clips. Although these methods capture long-range context, they introduce irrelevant negative samples, weakening frame-level associations. In contrast, our chain-structured contrastive formulation effectively balances local discriminability and long-range temporal consistency, significantly improving scalability and generalization.

**Open-world and scalable MOT.** The introduction of the TAO dataset Dave et al. (2020), containing over 800 classes, has spurred research in open-vocabulary MOT Zheng et al. (2024); Li et al. (2024a; 2022; 2023); Liu et al. (2022), emphasizing the ability of trackers to generalize beyond seen classes. Earlier benchmarks such as TAO-OW Liu et al. (2022) focused primarily on recall in class-agnostic scenarios, limiting insights into tracking precision and semantic understanding. OVTrack Li et al. (2023) broadened evaluation to include precision, recall, and classification accuracy, thereby better assessing trackers' semantic capabilities. Recently, MASA Li et al. (2024a) demonstrated the effectiveness of universal appearance models trained from unlabeled static images for zero-shot open-vocabulary tracking. However, these appearance-centric approaches generally neglect spatiotemporal and semantic context, crucial for robust tracking in diverse real-world scenarios. Transformer-based trackers Zeng et al. (2022); Meinhardt et al. (2022); Zhang et al. (2023b); Gao & Wang (2023); Segu et al. (2024) rely heavily on end-to-end sequence modeling, requiring exhaustive frame-level annotations for training. Their architecture lacks the flexibility to handle sparsely labeled data, making them unsuitable for scaling up for partially annotated video datasets. Unlike prior methods, our approach effectively leverages large-scale, sparsely annotated videos by introducing Chain Contrastive Learning, which balances local discriminability with long-range temporal consistency and demonstrates strong scaling capabilities.

## 3 METHOD

We introduce **ScaleTrack**, a transformer-based association framework for MOT, designed to leverage large-scale, sparsely annotated videos. Unlike conventional MOT architectures that rely on densely annotated training data, our approach effectively learns from partially labeled videos by combining a *Chain Contrastive Learning* strategy with pseudo-negative sampling and an efficient spatiotemporal attention module.

### 3.1 PROBLEM DEFINITION

Given a video sequence $\mathcal{V} = \{I_1, I_2, ..., I_T\}$, our goal is to assign a unique and consistent identity to each detected object across frames. Let $\mathcal{O}_t = \{o_t^1, o_t^2, ..., o_t^N\}$ represent the set of object instances detected at frame $t$, where each object $o_t^i$ is associated with a feature embedding $\mathbf{f}_t^i \in \mathbb{R}^d$ and a bounding box $\mathbf{b}_t^i \in \mathbb{R}^4$. The objective is to learn a function $f_\theta$ that extracts robust embeddings such that, for each object instance, its respective embedding is invariant across frames as well as distinct from those of other objects. ScaleTrack achieves this by (1) designing a *contrastive learning strategy*

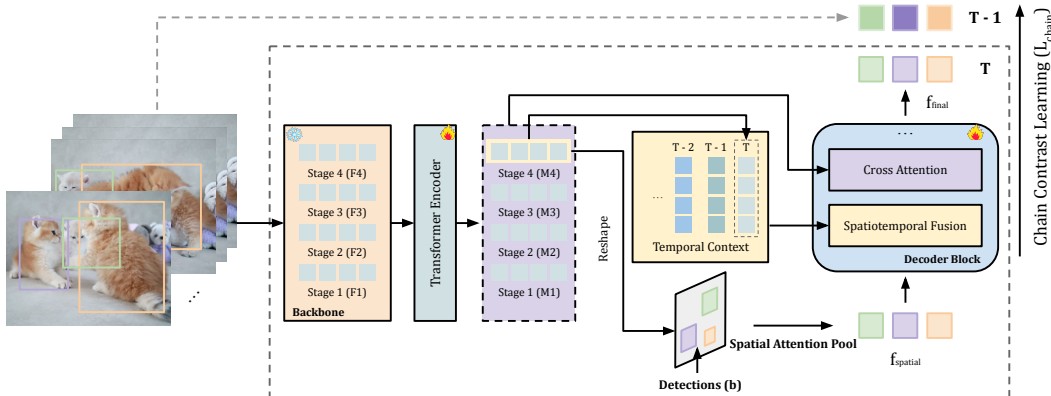

Figure 2: **Overview of our training framework.** Given an input video, our model extracts multi-scale features using a frozen visual backbone. These features are processed by a transformer encoder, which adapts them into a unified representation. The deepest feature level undergoes *spatial attention pooling* to enhance local object features. A *temporal memory buffer* maintains historical context, enabling our *spatiotemporal fusion module* in the decoder to aggregate relevant past information for robust object tracking. Finally, *cross-attention* refines object embeddings before association. The entire framework is optimized with *Chain Contrastive Learning*, ensuring both local discriminability and long-range temporal consistency.

that effectively scales to large video collections while preserving both local and long-term temporal consistency, (2) leveraging *pseudo-negative proposals* to improve feature discriminability, and (3) employing an efficient *spatiotemporal feature aggregation* module for robust object representation learning.

## 3.2 CHAIN CONTRASTIVE LEARNING

To overcome limitations of both purely local and global contrastive strategies, we propose *Chain Contrastive Learning*, a temporal contrastive paradigm specifically designed for robust multiple object tracking. The key idea is to construct contrastive chains of positive pairs across consecutive video frames, effectively balancing local discriminability with global temporal consistency.

Given a video clip consisting of a sequence of frames indexed by $t \in \{1, 2, \ldots, T\}$, let $\mathbf{f}_t^i \in \mathbb{R}^d$ represent the embedding of the $i$-th ground-truth object instance detected in frame $t$. We assume each annotated object maintains a consistent identity across the entire sampled clip. To enforce temporal consistency, we sample positive pairs of embeddings across consecutive frames in a chained manner: each instance in frame $t$ forms a positive pair with its counterpart in frame $t + 1$, resulting in chains of pairwise similarities through the entire clip.

More formally, given a pair of consecutive frames $(t, t+1)$, let $\mathcal{F}_t = \{\mathbf{f}_t^1, \mathbf{f}_t^2, \ldots, \mathbf{f}_t^{N_t}\}$ and $\mathcal{F}_{t+1} = \{\mathbf{f}_{t+1}^1, \mathbf{f}_{t+1}^2, \ldots, \mathbf{f}_{t+1}^{N_{t+1}}\}$ denote the sets of embeddings of ground-truth object instances in the two frames. Let $\mathcal{M}_{t,t+1}$ denote the set of matched, i.e. positive, instance pairs between these frames based on ground-truth identities.

To further enhance feature discriminability, particularly in scenarios with sparsely labeled data or to include challenging negative examples (as detailed in Sec. 3.3), we incorporate pseudo-negative proposals directly into our contrastive formulation. Let $\mathcal{P}_{t+1} = \{\mathbf{p}_{t+1}^1, \mathbf{p}_{t+1}^2, \ldots, \mathbf{p}_{t+1}^{M_{t+1}}\}$ represent the set of feature embeddings derived from these pseudo-negative proposals in frame $t + 1$.

The pairwise contrastive loss between two matched embeddings $(\mathbf{f}_t^i, \mathbf{f}_{t+1}^j)$ is then computed as the normalized temperature-scaled cross-entropy loss Sohn (2016), which contrasts the positive pair against both other ground-truth objects and the pseudo-negative proposals:

$$\mathcal{L}_{\text{pair}}(\mathbf{f}_t^i, \mathbf{f}_{t+1}^j) = -\log \frac{\exp(\mathbf{f}_t^i \cdot \mathbf{f}_{t+1}^j / \tau)}{\sum_{k \neq j} \exp(\mathbf{f}_t^i \cdot \mathbf{f}_{t+1}^k / \tau) + \sum_{m=1}^{M_{t+1}} \exp(\mathbf{f}_t^i \cdot \mathbf{p}_{t+1}^m / \tau) + \exp(\mathbf{f}_t^i \cdot \mathbf{f}_{t+1}^j / \tau)},$$

(1)

where $\tau$ is a temperature parameter. Negative examples for the anchor $\mathbf{f}_t^i$ are thus constructed from other ground-truth objects $\mathbf{f}_{t+1}^k$ (where $k \neq j$) in the neighboring frame $t+1$, as well as the set of pseudo-negative proposal embeddings $\mathbf{p}_{t+1}^m$ from the same frame $t+1$. This comprehensive set of negatives helps in learning highly discriminative features.

The final *Chain Contrastive Loss* for an entire video clip is computed by aggregating these pairwise losses across the sampled frame pairs:

$$\mathcal{L}_{\text{chain}} = \frac{1}{|\mathcal{M}|} \sum_{(t,t+1)\in\mathcal{T}} \sum_{(i,j)\in\mathcal{M}_{t,t+1}} \mathcal{L}_{\text{pair}}(\mathbf{f}_t^i, \mathbf{f}_{t+1}^j), \tag{2}$$

where $\mathcal{L}_{\text{pair}}$ is defined in Equation 1, $\mathcal{T}$ denotes the set of consecutive frame pairs, and $|\mathcal{M}|$ is the total number of positive pairs sampled across the entire video clip. This ensures that gradients propagate effectively through chained instances, implicitly enforcing long-range transitive consistency. More specifically, if object $A$ at time $t$ matches object $A'$ at $t+1$, and $A'$ matches $A''$ at $t+2$, then ScaleTrack transitively encourages similarity between $A$ and $A''$.

During training, embeddings are learned by minimizing the chain contrastive loss $\mathcal{L}_{\text{chain}}$ through standard backpropagation, thus enabling robust representation learning from video data.

### 3.3 Pseudo-Negative Proposals

To address challenges from sparsely annotated video data (e.g., in the SA-V dataset) and to enrich negative sampling with hard examples, we employ *pseudo-negative proposals*. These are unannotated, object-like region proposals ($\mathcal{R}_t = \{r_t^1, r_t^2, \ldots, r_t^{M_t}\}$ per frame) generated by applying large-scale foundation models, such as SAM Kirillov et al. (2023) and Detic Zhou et al. (2022), to each frame. Feature embeddings, denoted as $\mathbf{p}_t^m \in \mathbb{R}^d$, are extracted for each of these proposals.

As detailed in Section 3.2 (Equation 1), the embeddings of these proposals from frame $t+1$ (i.e., $\mathbf{p}_{t+1}^m$) directly augment the pool of negative samples in our pairwise Chain Contrastive Loss. Incorporating these diverse, hard pseudo-negatives alongside other ground-truth negatives significantly enhances feature discriminability and robustness.

### 3.4 Architecture

ScaleTrack, as shown in Fig. 2, consists of a transformer-based architecture with a spatiotemporal design for efficient object tracking. The framework consists of three main components: (1) a vision transformer backbone for multi-scale feature extraction, (2) a deformable transformer-based module for learning temporal dependencies and refining object representations, and (3) a tracking head that performs object association using contrastive learning.

#### 3.4.1 Transformer Encoder as Feature Adapter

To leverage frozen visual backbones (e.g., DINOv2 Oquab et al. (2024)), we use a lightweight deformable transformer encoder Zhu et al. (2020) as an adapter over *multi-level* features $\{\mathbf{F}_\ell\}_{\ell=1}^L$, where $\mathbf{F}_\ell \in \mathbb{R}^{B \times H_\ell \times W_\ell \times C_\ell}$. Each level is projected and encoded independently to

$$\mathbf{M}_\ell = \text{Encoder}_\ell(\mathbf{F}_\ell) \in \mathbb{R}^{B \times S_\ell \times D}, \quad S_\ell = H_\ell W_\ell,$$

then concatenated along the spatial axis:

$$\mathbf{M} = \text{Concat}_{\ell=1}^L \mathbf{M}_\ell \ \in \ \mathbb{R}^{B \times S \times D}, \quad S = \sum_{\ell=1}^L S_\ell. \tag{3}$$

Thus, $S$ denotes the total number of spatial locations across *all* feature levels via concatenation.

#### 3.4.2 Spatio-temporal Attention-Based Decoder

Our decoder refines object embeddings by integrating rich spatial and temporal context, which is crucial for accurate tracking in complex scenarios. Given proposal embeddings and reference points generated from the encoder output, the decoder enhances each proposal by performing two sequential attention-based steps: spatial attention pooling and temporal context fusion.

**Spatial attention pooling.** For each proposal, we first extract spatially aligned features applying RoIAlign He et al. (2017) on the final encoder features:

$$\mathbf{f}_{\text{RoI}} = \text{RoIAlign}(\mathbf{M}_L, \mathbf{b}), \tag{4}$$

where $\mathbf{M}_L$ denotes the final-level encoder features, and $\mathbf{b}$ represents the bounding box coordinates A small conv/MLP head predicts $K$ 2D offsets $\{(\Delta x_k, \Delta y_k)\}_{k=1}^{K}$ from $\mathbf{f}_{\text{RoI}}$. We apply `tanh` to constrain offsets to $[-1, 1]$ in the RoI coordinate system and sample at these locations via differentiable bilinear `grid_sample`, yielding $\mathbf{f}_{\text{sampled}}$. This guarantees sampling remains within the RoI and adds salient sub-region evidence. To enrich spatial context, we introduce a learnable spatial attention pooling module. This module computes attention weights over spatial locations within each RoI:

$$\mathbf{f}_{\text{att}} = \sum_{k,l} \mathbf{f}_{k,l} \cdot \sigma(\text{MLP}(\mathbf{f}_{k,l})), \tag{5}$$

where $\mathbf{f}_{x,y}$ are RoI features, and $\sigma$ denotes a spatial softmax. Concurrently, we introduce an adaptive sampling module that predicts learnable spatial offsets from RoI features. These offsets specify additional sampling locations within each RoI, from which contextually rich features are sampled using a differentiable bilinear sampling operation, denoted as $\mathbf{f}_{\text{sampled}}$. The spatial-attention-pooled features and the adaptively sampled features are then concatenated and fused via a linear layer, producing the final spatially enhanced embeddings:

$$\mathbf{f}_{\text{spatial}} = \text{Linear}\left([\mathbf{f}_{\text{attn}}, \mathbf{f}_{\text{sampled}}]\right). \tag{6}$$

**Temporal context fusion.** To incorporate temporal information, we maintain a buffer of historical feature maps extracted from the encoder, denoted as $\mathbf{F}_{\text{temp}} \in \mathbb{R}^{B \times D \times T \times H \times W}$, where $T$ is the temporal window size. Inspired by deformable attention mechanisms, our temporal fusion module dynamically samples relevant spatiotemporal features from this buffer. Specifically, for each object proposal embedding $\mathbf{f}_{\text{spatial}} \in \mathbb{R}^{B \times L \times D}$, we predict adaptive sampling offsets $(\Delta t, \Delta x, \Delta y)$ and corresponding attention weights. These offsets guide a deformable sampling procedure, which extracts informative features from neighboring spatial and temporal locations within $\mathbf{F}_{\text{temp}}$. The sampled features are then aggregated using learned attention weights to produce the temporally-enhanced representation:

$$\mathbf{f}_{\text{fused}} = \text{SpatiotemporalFusion}(\mathbf{f}_{\text{spatial}}, \mathbf{F}_{\text{temp}}). \tag{7}$$

This adaptive sampling strategy enables our model to selectively incorporate relevant historical context, effectively capturing both local spatial detail and long-range temporal coherence. Importantly, due to its sparse sampling nature, this approach remains computationally efficient.

*3D deformable fusion.* Given $\mathbf{f}_{\text{spatial}}$, a linear head predicts $H \times P$ 3D offsets $\{(\Delta t, \Delta x, \Delta y)\}$ and attention weights $\{\alpha\}$ over the cached volume $\mathbf{F}_{\text{temp}}$. Offsets are `tanh`-bounded and scaled to the temporal window; weights are softmax-normalized. We then perform trilinear `grid_sample` in $(t, x, y)$ and aggregate by $\alpha$ to obtain $\mathbf{f}_{\text{fused}}$ (Eq. 8), efficiently injecting long-range history.

**Proposal attention modulation.** To further improve discriminability, proposal embeddings undergo head-specific feature modulation through learned scaling factors:

$$\mathbf{f}_{\text{modulated}} = \frac{1}{N_h} \sum_{h=1}^{N_h} \sigma(\mathbf{W}_h \mathbf{f}_{\text{fused}}) \cdot \mathbf{f}_{\text{fused}}, \tag{8}$$

where $N_h$ is the number of attention heads, and $\mathbf{W}_h$ denotes head-specific learnable linear transformations.

**Cross-attention refinement.** The spatially and temporally enhanced embeddings serve as queries for a multi-scale cross-attention module, aggregating rich spatial context from encoder memory across multiple feature scales of the current frame:

$$\mathbf{f}_{\text{final}} = \text{CA}(\mathbf{f}_{\text{modulated}}, \mathbf{M}), \tag{9}$$

where $\mathbf{M}$ contains multi-level spatial features. This aggregation captures detailed spatial relationships within the current frame, producing robust, contextually enriched object embeddings crucial for accurate object tracking.

Table 1: **Comparison with SOTAs on Open-vocabulary MOT.** †: same detector.

| Method | Novel | | | | Base | | | |
|---|---|---|---|---|---|---|---|---|
| Validation set | TETA | LocA | AssocA | ClsA | TETA | LocA | AssocA | ClsA |
| *Supervised, In-domain* | | | | | | | | |
| QDTrack Pang et al. (2021) | 22.5 | 42.7 | 24.4 | 0.4 | 27.1 | 45.6 | 24.7 | 11.0 |
| TETer Li et al. (2022) | 25.7 | 45.9 | 31.1 | 0.2 | 30.3 | 47.4 | 31.6 | 12.1 |
| DeepSORT (ViLD) Wojke et al. (2017) | 21.1 | 46.4 | 14.7 | 2.3 | 26.9 | 47.1 | 15.8 | 17.7 |
| Tractor++ (ViLD) Bergmann et al. (2019) | 22.7 | 46.7 | 19.3 | 2.2 | 28.3 | 47.4 | 20.5 | 17.0 |
| ByteTrack Zhang et al. (2022) | 22.0 | 48.2 | 16.6 | 1.0 | 28.2 | 50.4 | 18.1 | 16.0 |
| OC-SORT Cao et al. (2023) | 23.7 | 49.6 | 20.4 | 1.1 | 28.9 | 51.4 | 19.8 | 15.4 |
| OVTrack Li et al. (2023) | 27.8 | 48.8 | 33.6 | 1.5 | 35.5 | 49.3 | 36.9 | 20.2 |
| SLAck Li et al. (2024b) | 31.1 | 54.3 | 37.8 | 1.3 | 37.2 | 55.0 | 37.6 | 19.1 |
| *Zero-shot* | | | | | | | | |
| NetTrack Zheng et al. (2024) | 32.6 | 51.3 | 33.0 | 13.3 | 33.0 | 45.7 | 28.6 | 24.8 |
| MASA (Detic)† Li et al. (2024a) | 40.8 | 64.4 | 41.2 | 17.0 | 47.0 | 66.0 | 44.5 | 30.5 |
| **ScaleTrack** (Detic)† | 44.0 | 65.8 | 48.6 | 17.7 | 49.1 | 66.3 | 52.3 | 28.7 |

Table 2: **Comparison with SOTAs on BDD MOT(val set).** †: same detector.

| Method | TETA↑ | LocA↑ | AssocA↑ | mIDF1↑ | IDF1↑ | mHOTA↑ |
|---|---|---|---|---|---|---|
| *Supervised, in-domain* | | | | | | |
| QDTrack Pang et al. (2021) | 47.8 | 45.9 | 52.9 | 51.6 | 71.5 | - |
| TETer Li et al. (2022) | 50.8 | 47.2 | 52.9 | 53.3 | 71.1 | - |
| MOTR Zeng et al. (2022) | 50.7 | 35.8 | 51.0 | 54.0 | 65.8 | - |
| UNINEXT-H Yan et al. (2023) | - | - | - | 56.7 | 69.9 | - |
| ByteTrack Zhang et al. (2022)† | 55.7 | - | 51.5 | 54.8 | 70.4 | - |
| MeMOTR Gao & Wang (2023) | 53.6 | 38.1 | 56.7 | 48.8 | 69.2 | 40.4 |
| *Zero-shot* | | | | | | |
| MASA (SAM-H)† | 54.2 | 53.7 | 51.9 | 55.3 | 71.7 | 45.9 |
| MASA (Detic)† | 54.4 | 52.7 | 52.9 | 55.8 | 71.3 | 46.2 |
| **ScaleTrack†** | 56.3 | 53.9 | 56.0 | 56.7 | 73.8 | 46.6 |

## 3.5 INFERENCE

During inference, ScaleTrack extracts object embeddings and associates detections to existing tracks using pairwise cosine similarity matching. The detailed assignment process, which utilizes the Hungarian algorithm, is described in Appendix Sec. A.

## 4 EXPERIMENTS

### 4.1 EXPERIMENTAL SETUP

**Implementation details.** We train ScaleTrack for 5 epochs using the AdamW optimizer Loshchilov & Hutter (2018), with an initial learning rate of $10^{-4}$, decayed after 2 epochs. The model is trained with a batch size of 1 video clip and 10 frames per clip. We set the transformer hidden dimension to 256. We add more details in the appendix.

**Evaluation metrics.** We evaluate large-scale multi-class MOT using the TETA metrics Li et al. (2022). TETA decomposes tracking evaluation into three aspects: Localization (LocA), Association (AssocA), and Classification (ClsA), providing an interpretable analysis of tracker performance. Moreover, TETA accommodates incomplete annotations, particularly suiting open-vocabulary MOT Li et al. (2023). For standard MOT benchmarks, we ensure a fair comparison of association performance by using the same detections as leading trackers. Consequently, our primary focus is on *association-related* metrics, including AssocA, mIDF1, and IDF1.

**Benchmarks.** We evaluate our method on multiple datasets, each presenting unique challenges in MOT. **OV-MOT Li et al. (2023)** assesses a tracker's ability to generalize to novel object classes without training on their annotations. TAO follows the LVIS Gupta et al. (2019) taxonomy, where frequent and common classes form the base set, while rare ones constitute the novel set. This split aligns with open-vocabulary detection protocols Gu et al. (2021), ensuring that trackers must adapt to unfamiliar categories, a critical requirement for real-world applications. **TAO TETA Li et al. (2022)** is a closed-set MOT benchmark allowing training on all class annotations within TAO. TAO Dave et al. (2020) is designed to track a diverse range of objects, encompassing over 800 categories, making it the most diverse MOT dataset with the largest class collection to date. It contains 500, 988, and 1,419 videos annotated at 1 FPS in the train, validation, and test sets, respectively. We report performance on the validation set. TAO emphasizes the quality of association, rewarding trackers that generate accurate and non-overlapping trajectories. **BDD100K MOT Yu et al. (2020)** evaluates tracking performance in driving scenes. It includes 200 validation videos annotated at 5 FPS, requiring trackers to handle common traffic objects in dynamic urban environments. **SportsMOT Cui et al. (2023)** is a large-scale MOT benchmark of sports scenes, covering basketball, volleyball, and football. It consists of 240 videos, 150K frames, and 1.6M bounding boxes, and presents challenges such as fast, variable-speed motion and visually similar players, making object association a key challenge. **OVT-B Liang & Han (2025)** is a large-scale open-vocabulary MOT benchmark. The detailed results can be found in the Appendix Sec. C.

**Training data.** The goal of this work is to scale up MOT training using sparsely annotated video data. To study the effects of large-scale training, we leverage the **SA-V** dataset Ravi et al. (2024), a large-scale video object segmentation dataset with 35.5M masks across 50.9K videos. Unlike traditional datasets, SA-V provides dense mask annotations for a diverse range of objects, including parts and subparts, making it well-suited for open-world tracking. Our final model is trained on subsampled sequences from `sav_000` to `sav_023`.

Table 3: **Comparison with the state-of-the-art on TAO TETA** [†]: usage of the same detector.

| Method | TETA | LocA | AssocA | ClsA |
|---|---|---|---|---|
| *Supervised, in-domain* | | | | |
| SORT Bewley et al. (2016) | 24.9 | 48.1 | 14.3 | 12.1 |
| Tracktor Bergmann et al. (2019) | 24.2 | 47.4 | 13.0 | 12.1 |
| Tracktor++ Bergmann et al. (2019) | 28.0 | 49.0 | 22.8 | 12.1 |
| DeepSORT Wojke et al. (2017) | 26.0 | 48.4 | 17.5 | 12.1 |
| QDTrack Pang et al. (2021) | 30.0 | 50.5 | 27.4 | 12.1 |
| TETer Li et al. (2022) | 34.6 | 52.1 | 36.7 | 15.0 |
| SLAck-T[†] Li et al. (2024b) | 35.5 | 52.2 | 38.9 | 15.6 |
| SLAck-L[†] Li et al. (2024b) | 41.1 | 56.3 | 41.8 | 25.1 |
| *Zero-shot* | | | | |
| UNINEXT(R50) Yan et al. (2023) | 31.9 | 43.3 | 35.5 | 17.1 |
| GLEE-Plus(SwinL) Wu et al. (2023) | 41.5 | 52.9 | 40.9 | 30.8 |
| GLEE-Pro(EVA02-L) Wu et al. (2023) | 47.2 | 66.2 | 46.2 | 29.1 |
| MASA (Detic) [†] Li et al. (2024a) | 46.3 | 65.8 | 44.1 | 28.9 |
| **ScaleTrack** (Detic)[†] | 48.5 | 66.2 | 51.9 | 27.4 |
| **ScaleTrack** (CoDETR) | **57.7** | **71.6** | **60.3** | **41.6** |

Table 4: **Comparison with the state-of-the-art on SportsMOT (test set).** We use official SportsMOT baseline detections. [†]: usage of the same detection observations.

| Methods | HOTA | AssA | DetA | IDF1 | MOTA |
|---|---|---|---|---|---|
| *Supervised, in-domain* | | | | | |
| QDTrack Pang et al. (2021) | 60.4 | 47.2 | 77.5 | 62.3 | 90.1 |
| ByteTrack Zhang et al. (2022) | 62.1 | 50.5 | 76.5 | 69.1 | 93.4 |
| OC-SORT Cao et al. (2023) | 68.1 | 54.8 | 84.8 | 68.0 | 93.4 |
| TransTrack Sun et al. (2020b) | 68.9 | 57.5 | 82.7 | 71.5 | 92.6 |
| MeMOTR Gao & Wang (2023) | 68.8 | 57.8 | 82.0 | 69.9 | 90.2 |
| SambaMOTR Segu et al. (2024) | 69.8 | 59.4 | 82.2 | 71.9 | 90.3 |
| MOTIP Gao et al. (2024) | 71.9 | 62.0 | 83.4 | 75 | 92.9 |
| MixSort-Byte Cui et al. (2023)[†] | 65.7 | 54.8 | 78.8 | 74.1 | 96.2 |
| MixSort-OC Cui et al. (2023)[†] | 74.1 | 62.0 | **88.5** | 74.4 | **96.5** |
| *Zero-shot* | | | | | |
| **ScaleTrack**[†] | **76.2** | **66.7** | 87.0 | **79.3** | 95.0 |

Table 5: **Scaling ablation on chain length.**

| Chain Length | 3 | 5 | 10 | 20 |
|---|---|---|---|---|
| AssocA | 52.8 | 54.4 | 54.7 | 55.8 |

Table 6: **Scaling ablation on backbone.**

| Backbone | 3k | 6k | 12k |
|---|---|---|---|
| DINOv2-S (Frozen) | 57.5 | 57.7 | 58.6 |
| DINOv2-S (Unfrozen) | 55.9 | 57.8 | 59.5 |

Table 7: **Ablation of model components.**

| Frozen DINOv2 | Chain Contrast | Spatial Att Pool | Cross Att | Temporal Att | AssocA |
|---|---|---|---|---|---|
| ✓ | | | | | 47.9 |
| ✓ | ✓ | | | | 55.2 |
| ✓ | ✓ | ✓ | | | 55.7 |
| ✓ | ✓ | ✓ | ✓ | | 56.7 |
| ✓ | ✓ | ✓ | ✓ | ✓ | 57.3 |

## 4.2 COMPARISON WITH THE STATE-OF-THE-ART

To accurately assess the association ability of our method, we always provide the same detection observations as current state-of-the-art methods in standard MOT benchmarks. Note that we perform zero-shot association tests for all our variants and *use the same weights* across all benchmarks.

**OV-MOT.** Tab. 1 evaluates trackers on the Open-Vocabulary MOT benchmark, which assesses generalization to novel categories following the LVIS Gupta et al. (2019) taxonomy. Supervised in-domain methods, such as SLAck and OVTrack, perform well but require extensive labeled data, limiting scalability. In the *zero-shot* setting, our **ScaleTrack** outperforms the prior state-of-the-art, surpassing MASA by 3.2% in TETA and 7.4% in AssocA on novel classes, demonstrating superior object association.

**TAO TETA.** We use the same observations as MASA Detic Li et al. (2024a). As shown in Tab. 3, our method performs the best, without training on any in-domain data, on both AssocA and TETA by a wide margin and sets the new state-of-the-art. This demonstrates that our method couples well with current detection foundation models and transfers their strong detection performance into tracking.

**BDD100K MOT.** Tab. 2 shows that among supervised trackers on BDD100K, MeMOTR Gao & Wang (2023) achieves strong AssocA performance but requires dataset-specific tuning. In the *zero-shot* setting, **ScaleTrack** surpasses MASA, achieving the best TETA (+1.9%) and AssocA (+3.1%), along with the highest IDF1 and HOTA. Our results demonstrate that ScaleTrack achieves state-of-the-art generalization in real-world driving environments.

**SportsMOT.** Tab. 4 presents results on the SportsMOT benchmark, which poses significant challenges due to fast motion and visually similar players. In the *zero-shot* setting, **ScaleTrack** outperforms all prior methods, including supervised trackers, achieving the highest HOTA (+2.1% from MixSort-OC) and IDF1 (+4.7%). These results demonstrate the effectiveness of our proposed contrastive ScaleTrack training and our spatiotemporal attention in handling fine-grained object association.

## 4.3 ABLATION STUDIES AND ANALYSIS

We conduct all ablation studies on the TAO dataset using Co-DETR Zong et al. (2023) as the default detector. Unless otherwise specified, model variants are trained on 1k videos. For scaling experiments, including loss scaling, clip length scaling, and frozen feature scaling, we train each model for a single epoch, as we found that one epoch provides sufficiently informative trends for comparison. For model ablation, we train for 5 epochs. Below, we analyze the impact of different design choices in our framework.

**Scaling effect of different contrastive strategies.** To evaluate the scalability of different contrastive strategies, we trained all methods, our chain contrast, alongside Local Pairwise, Globe contrast with

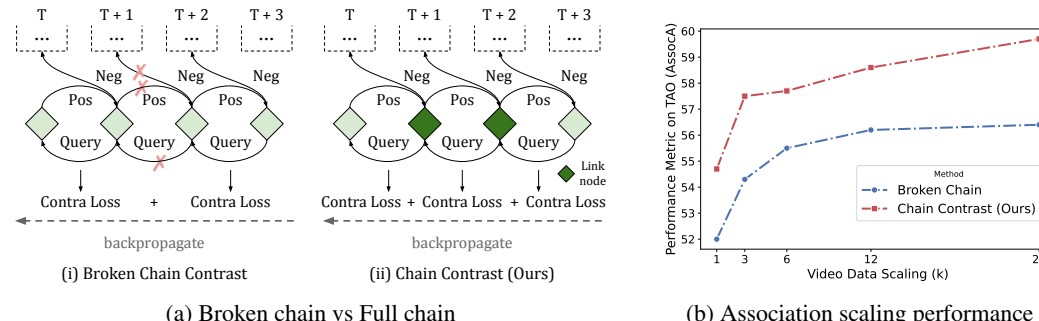

(a) Broken chain vs Full chain      (b) Association scaling performance

Figure 3: **Ablation of chain structure in contrastive learning for tracking.** (a) **Concept: *Broken Chain* vs. *Chain Contrast*.** (i) The *Broken Chain* intentionally disrupts inter-frame positive links, impeding transitive consistency. (ii) Our *Chain Contrast* maintains these links to enable long-range feature propagation. (b) **Performance scaling: *Chain Contrast (Ours)* vs. *Broken Chain*.** As video training data increases, *Chain Contrast (Ours)* (red line) significantly outperforms the *Broken Chain* (blue line) in association accuracy (AssocA) on the TAO dataset. This highlights the critical role of the continuous chain structure for robust long-range consistency and tracking scalability.

SupCon and InfoNCE baselines on the same model architecture and data. This controlled setting allows for a direct comparison of how effectively each loss function leverages increasing amounts of data. Fig. 1b demonstrates that our proposed chain contrast scales the best.

**Scaling effect of chain structure.** We investigate the impact of the proposed *chain* structure in our contrastive learning framework. As shown in Fig. 3, breaking the chain structure (Broken Chain) by independently optimizing contrastive pairs between isolated frame-pairs significantly degrades performance, indicating that the chain structure is crucial for propagating meaningful long-range associations.

**Scaling effect of chain length.** Tab. 5 evaluates how varying the temporal length of the contrastive chains affects tracking performance. Increasing the chain length consistently enhances association accuracy, highlighting the importance of leveraging longer temporal context.

**Scaling effect of frozen vs. unfrozen backbone.** We present an ablation on freezing the DINOv2 backbone in Tab. 6. With smaller datasets (3k), the frozen backbone performs better due to DINOv2's pretraining. As training data increases (12k+), the unfrozen backbone better adapts to task-specific patterns. While fine-tuning provides minor improvements at larger scales, freezing the backbone remains computationally efficient and preserves the diversity of representations learned from large-scale pre-training.

**Effectiveness of model components.** Tab. 7 shows the impact of individual model components on TAO. Starting from a frozen DINOv2 backbone, adding the Transformer-based feature adapter and training with chained contrastive loss significantly improves performance by adapting static image features for tracking. Incorporating Spatial Attention Pooling further enriches spatial context, yielding an additional gain. Adding Cross Attention effectively aggregates multi-level spatial features within the current frame, further enhancing object discrimination. Finally, including Temporal Fusion, which explicitly integrates historical context, achieves the best overall result.

## 5 CONCLUSION

We introduced **ScaleTrack**, a novel transformer-based association framework that effectively leverages large-scale, sparsely annotated video data for robust and generalizable tracking. Our proposed *Chain Contrastive Learning* not only balances local discriminability with long-range temporal consistency, outperforming traditional local and global contrastive paradigms, but also scales effectively with increasing training data. Additionally, leveraging pseudo-negative proposals enriches training with hard negatives, further enhancing discriminative capability. Extensive experiments demonstrate that ScaleTrack achieves state-of-the-art zero-shot performance across diverse MOT benchmarks, highlighting its scalability and suitability for open-world tracking scenarios.

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

APPENDIX

In this appendix, we report additional discussions and experiments. First, we provide implementation details in Sec. A. Then, we report additional quantitative results on more benchmarks in Sec.B and Sec. C. Finally, we provide additional ablations on the specifics of our method in Sec. D.

## A  IMPLEMENTATION DETAILS

We provide more implementation details here including training, testing, and model architecture.

### A.1  TRAINING

We train ScaleTrack for 5 epochs using AdamW optimizer Loshchilov & Hutter (2018) with initial learning rate of $2 \times 10^{-4}$, decayed after epoch 2. The backbone learning rate is set to $2 \times 10^{-5}$, while linear projection layers use the same learning rate as the transformer. We employ gradient clipping with a maximum norm of 10. The training batch size is 1 video clip, with each clip consisting of 10 uniformly sampled frames. For data augmentation, we apply Large-Scale Jittering (LSJ).

**Model architecture.**  ScaleTrack uses a frozen DINOv2-S Oquab et al. (2024) backbone, followed by a 2-layer deformable transformer encoder. We also have 2 layers of our proposed decoder, each with a hidden dimension of 256 and 4 sampling points per attention head. We incorporate a spatial attention pooling module and temporal fusion to aggregate rich spatiotemporal context.

**Data and annotations.**  All models are trained on the SA-V dataset Ravi et al. (2024), using object proposals generated by Detic. During training, we utilize pseudo-labels from Detic Zhou et al. (2022) predictions to augment sparse annotations. Training and validation detection confidence thresholds are set to 0.3, with validation Non-Maximum Suppression (NMS) at 0.4 IoU.

**Losses and matching.**  We use chain contrastive loss to supervise association. Object tracks are matched using Hungarian matching, with an association threshold of 0.5. Embeddings for matching are learned via our chain contrastive strategy.

**Inference.**  During inference, ScaleTrack extracts object embeddings and performs association via a simple yet effective similarity-based matching strategy. Given extracted embeddings, we compute pairwise cosine similarity scores between detected objects and existing tracks. Object associations are determined using the Hungarian algorithm, assigning detections to tracks by minimizing the cost:

$$\text{HungarianMatching}(1 - \mathbf{S}, \tau), \tag{10}$$

where $\mathbf{S}$ is the similarity matrix, and $\tau$ is a threshold to filter low-confidence matches. We retain up to 100 detections per frame at inference. Track states are updated through an exponential moving average (EMA) with a weight of 0.5.

**Efficiency.**  To enhance computational efficiency, we freeze backbone parameters and apply automatic mixed precision (AMP). With these optimizations, ScaleTrack achieves 42 FPS on an NVIDIA RTX 4090 GPU.

**Resources.**  All ScaleTrack variants and ablation experiments were implemented in PyTorch and trained on a cluster equipped with NVIDIA H100 GPUs (typically 96GB memory versions). For our main model configurations, training was performed using data parallelism across 16 H100 GPUs. A typical training run for 5 epochs on a substantial subset of the SA-V dataset (e.g., 24k video clips) with a per-GPU batch size of 1 video clip, each clip containing 10 to 20 frames, took approximately 3 days. As noted in the abstract, inference is performed on a single NVIDIA RTX 4090 GPU.

## B  IMPACT OF LARGE-SCALE PRE-TRAINING VS. IN-DOMAIN TRAINING

A core objective of ScaleTrack is to effectively leverage large-scale, sparsely annotated video data, and our primary models are trained on the SA-V dataset to demonstrate this capability. While larger

and more diverse datasets generally contribute to improved generalization, a valid question is the extent to which ScaleTrack's strong performance relies solely on the scale of SA-V versus the inherent strengths of our architecture and Chain Contrastive Learning approach.

To investigate this, we conducted an additional experiment on the SportsMOT benchmark Cui et al. (2023). For this experiment, we trained ScaleTrack **exclusively** with the official SportsMOT training set, using the same in-domain data as other comparable supervised methods. This ensures a direct comparison of model capabilities when data conditions are equalized, removing the advantage of large-scale pre-training on SA-V.

Table 8: **Performance on SportsMOT with In-Domain Training.** Comparison of ScaleTrack trained only on SportsMOT data against methods using the same training set. Metrics like HOTA and IDF1 are shown.

| Method | Training Data | HOTA ↑ | AssA ↑ | IDF1 ↑ |
|---|---|---|---|---|
| SambaMOTR Segu et al. (2024) | SportsMOT | 69.8 | 59.4 | 71.9 |
| MixSort-OC Cui et al. (2023) | SportsMOT | 74.1 | 62.0 | 74.4 |
| **ScaleTrack (Ours)** | **SA-V** | 76.2 | 66.7 | 79.3 |
| **ScaleTrack (Ours)** | **SportsMOT** | **76.8** | **67.7** | **80.1** |

As shown in Tab. 8, even when trained solely on the in-domain SportsMOT data, ScaleTrack achieves a HOTA of 76.8 and an IDF1 of 80.1, surpassing previous SOTA by a large margin.

This result demonstrates that while ScaleTrack benefits significantly from large-scale data like SA-V for broad generalization (as shown in our main results), the core architectural design and the Chain Contrastive Learning strategy provide substantial advantages even in standard, in-domain training scenarios. It underscores that the performance gains are not solely attributable to the volume of pre-training data but also to the effectiveness of our proposed tracking framework in learning discriminative and robust object associations.

## C  MORE BENCHMARKS

We report additional results on the OVT-B Liang & Han (2025) benchmark and its individual subsets.

**OVT-B Liang & Han (2025)** is a large-scale open-vocabulary MOT benchmark with 1,048 object categories and 1,973 videos containing 637,608 bounding box annotations. Built from seven datasets—AnimalTrack Zhang et al. (2023a), GMOT-40 Bai et al. (2021), LV-VIS Wang et al. (2023), OVIS Qi et al. (2022), UVO Wang et al. (2021a), YouTube-VIS Xu et al. (2018), and ImageNet-VID Deng et al. (2009), it introduces diverse challenges, including severe occlusions (OVIS) and small, visually similar objects (GMOT-40). OVT-B serves exclusively as a challenging evaluation benchmark for open-vocabulary tracking.

To provide a more detailed analysis of our model's generalization ability, we also report results on individual OVT-B subsets, including UVOWang et al. (2021a), OVISQi et al. (2022), YouTube-VISXu et al. (2018), AnimalTrack Zhang et al. (2023a), GMOT-40Bai et al. (2021), ImageNetVIDDeng et al. (2009), and LV-VIS Wang et al. (2023). These subsets span various tracking difficulties, including occlusion, scale variation, motion complexity, and long-tail category distributions.

- **UVO** (*Unidentified Video Objects*): Focuses on open-world object tracking, with a vast vocabulary and many unlabeled background objects.
- **OVIS** (*Occluded Video Instance Segmentation*): Features severe occlusions, making object association particularly difficult.
- **YouTube-VIS**: A standard benchmark for video instance segmentation, containing diverse objects in dynamic real-world scenes.
- **AnimalTrack**: Contains multiple species in natural environments, posing challenges with non-rigid deformations and varying motion patterns.
- **GMOT-40**: A general multi-object tracking dataset with small, visually similar objects, requiring precise association across frames.

Table 9: **Comparison with the state of the art on OVT-B Liang & Han (2025).** We compare our method on open-vocabulary MOT benchmark Li et al. (2023). We indicate the classes and data the methods are trained on. $^{\dagger}$: usage of the same detector.

| | All | | | | Base | | | | Novel | | | |
|---|---|---|---|---|---|---|---|---|---|---|---|---|
| Method | TETA | LocA | AssocA | ClsA | TETA | LocA | AssocA | ClsA | TETA | LocA | AssocA | ClsA |
| ByteTrack Zhang et al. (2022) | 20.1 | 36.1 | 12.4 | 11.9 | 20.6 | 35.6 | 12.7 | 13.4 | 19.6 | 36.6 | 12.0 | 10.3 |
| OC-SORT  Cao et al. (2023) | 16.0 | 31.2 | 4.3 | 12.3 | 16.5 | 31.0 | 4.4 | 14.3 | 15.4 | 31.4 | 4.3 | 10.3 |
| StrongSORT Du et al. (2023) | 24.8 | 31.6 | 30.7 | 12.2 | 25.7 | 31.4 | 31.6 | 14.2 | 23.9 | 31.8 | 29.7 | 10.3 |
| OVTrack Li et al. (2023) | 46.1 | 60.8 | 66.1 | 11.5 | 46.8 | 60.5 | 66.7 | 13.4 | 45.5 | 61.1 | 65.5 | 9.6 |
| OVTrack+ Liang & Han (2025) | 47.0 | 62.0 | 67.7 | 11.3 | 47.6 | 61.6 | 68.2 | 13.2 | 46.4 | 62.5 | 67.3 | 9.4 |
| MASA (Detic) Li et al. (2024a)$^{\dagger}$ | 57.4 | **73.0** | 77.7 | 20.6 | 59.2 | **73.2** | 78.1 | 26.2 | 55.5 | **74.1** | 77.3 | 15.0 |
| ScaleTrack (Detic) $^{\dagger}$ | **57.8** | 72.8 | **79.9** | **21.0** | **59.6** | 72.5 | **80.0** | **26.5** | **55.9** | 73.0 | **79.8** | 15.0 |

Table 10: **Comparison with the state of the art across multiple datasets.**

(a) **OVIS**

| Method | TETA | LocA | AssocA | ClsA |
|---|---|---|---|---|
| ByteTrack | 31.5 | 49.0 | 3.4 | 42.1 |
| OVTrack | 48.4 | 58.5 | 41.0 | **45.7** |
| MASA | 54.9 | **73.7** | 51.5 | 39.6 |
| **ScaleTrack** | **55.9** | 72.1 | **55.2** | 40.3 |

(b) **LVVIS**

| Method | TETA | LocA | AssocA | ClsA |
|---|---|---|---|---|
| ByteTrack | 19.1 | 31.0 | 9.6 | 16.6 |
| OVTrack | 40.3 | 53.2 | 49.9 | 17.6 |
| MASA | 55.3 | **66.9** | 63.3 | **35.7** |
| **ScaleTrack** | **56.2** | **66.9** | **66.1** | 35.5 |

(c) **GMOT-40**

| Method | TETA | LocA | AssocA | ClsA |
|---|---|---|---|---|
| ByteTrack | 21.2 | 23.4 | 1.1 | 39.2 |
| OVTrack | 38.0 | 40.8 | 32.7 | 40.4 |
| MASA | **47.3** | **54.8** | 42.1 | **44.9** |
| **ScaleTrack** | 46.9 | 52.8 | **43.3** | 44.7 |

(d) **UVO**

| Method | TETA | LocA | AssocA | ClsA |
|---|---|---|---|---|
| ByteTrack | 11.4 | 11.3 | 3.4 | 19.5 |
| OVTrack | 25.6 | 27.9 | 26.4 | 22.4 |
| MASA | 32.3 | 33.5 | 31.7 | 31.7 |
| **ScaleTrack** | **33.6** | **34.5** | **32.2** | **34.2** |

(e) **YouTubeVIS**

| Method | TETA | LocA | AssocA | ClsA |
|---|---|---|---|---|
| ByteTrack | 14.8 | 29.5 | 4.1 | 10.8 |
| OVTrack | 36.9 | 54.8 | 43.3 | 12.7 |
| MASA | 47.7 | 62.7 | 51.7 | 28.7 |
| **ScaleTrack** | **57.0** | **68.8** | **61.5** | **40.7** |

(f) **AnimalTrack**

| Method | TETA | LocA | AssocA | ClsA |
|---|---|---|---|---|
| ByteTrack | 22.9 | 35.6 | 0.6 | 32.4 |
| OVTrack | 40.9 | 53.9 | 39.1 | 29.7 |
| MASA | **57.4** | 66.4 | **51.3** | 54.6 |
| **ScaleTrack** | 57.2 | **66.6** | 50.2 | **54.8** |

(g) **ImageNetVID**

| Method | TETA | LocA | AssocA | ClsA |
|---|---|---|---|---|
| ByteTrack | 21.6 | 37.1 | 1.2 | 26.5 |
| OVTrack | 42.2 | 55.0 | 43.4 | 28.2 |
| MASA | **55.4** | **70.6** | 58.5 | 37.1 |
| **ScaleTrack** | **55.4** | 68.8 | **60.1** | **37.2** |

- **ImageNetVID**: A subset of ImageNet focusing on object tracking, commonly used for evaluating appearance-based models.

- **LV-VIS** (*Large-Vocabulary Video Instance Segmentation*): Extends VIS with a broader category set, emphasizing tracking in a diverse, large-scale vocabulary.

## C.1 RESULTS ON OVT-B

Tab. 9 shows that **ScaleTrack** achieves the highest overall TETA and AssocA on one of the most diverse and challenging MOT benchmarks, i.e. OVT-B, demonstrating superior tracking robustness across highly diverse and occluded scenarios. Notably, our method surpasses OVTrack+ by 10.8%

in TETA and 12.2% in AssocA, highlighting the effectiveness of our scalable training approach on sparsely annotated large-scale video data.

## C.2 Results on OVIS

OVIS presents one of the most challenging multiple object tracking (MOT) scenarios due to its severe occlusions and highly dynamic object interactions. Tab. 10a compares our method, **ScaleTrack**, against state-of-the-art trackers on this benchmark.

ScaleTrack achieves the highest **AssocA** (55.2), outperforming MASA by **+3.7** points and OVTrack by **+14.2**. This significant improvement highlights the effectiveness of our *Chain Contrastive Learning* in ensuring long-range feature consistency while maintaining strong local discriminability. Unlike prior methods that struggle with severe occlusions, our approach better retains object identity across frames by leveraging transitive consistency in feature learning.

While MASA achieves the best **LocA** (73.7), our method follows closely at 72.1, showing that our robust spatiotemporal modeling does not compromise localization accuracy. Additionally, our model achieves the highest **TETA** (55.9), confirming its superior overall tracking performance in challenging occlusion-heavy environments.

## C.3 Results on LV-VIS

LV-VIS significantly expands the vocabulary size of standard VIS benchmarks, making it an excellent testbed for open-vocabulary tracking. As shown in Tab. 10b, ScaleTrack outperforms existing state-of-the-art methods in most metrics.

Our method achieves the highest **TETA** (56.2) and **AssocA** (66.1), demonstrating its strong ability to maintain identity consistency across a vast number of object categories. Compared to OVTrack, we improve **AssocA** by **+16.2** points, showcasing the superior scalability of our learned representations.

Compared to MASA, which leverages static-image-based contrastive learning, our approach achieves better **TETA** and **AssocA**, highlighting the importance of modeling temporal dynamics explicitly. While MASA attains the best **ClsA** (35.7), our model closely matches it (35.5), confirming our ability to track diverse objects while preserving their semantic identity.

## C.4 Results on GMOT-40

GMOT-40 presents a unique challenge in MOT, featuring numerous small and visually similar objects, making association difficult. As shown in Tab. 10c, ScaleTrack achieves the highest **AssocA** (43.3), surpassing MASA by **+1.2** and OVTrack by **+10.6**. This demonstrates that our approach effectively captures fine-grained object differences, leading to more robust tracking. Additionally, our method attains competitive **TETA** (46.9) and **ClsA** (44.7), highlighting its strong open-vocabulary tracking capabilities.

## C.5 Results on UVO

UVO is a large-scale dataset covering diverse, unlabeled objects, emphasizing open-world tracking. As shown in Tab. 10d, ScaleTrack consistently outperforms all baselines, achieving the best **TETA** (33.6) and **AssocA** (32.2). Compared to MASA, which is trained on large-scale image datasets, our approach better leverages spatiotemporal information, leading to a **+0.5** gain in **AssocA** and **+2.5** in **ClsA**. These results suggest that our method generalizes well to highly diverse real-world scenes.

## C.6 Results on YouTubeVIS

YouTubeVIS is a widely used benchmark for video instance segmentation, offering a rich variety of objects and motion patterns. As shown in Tab. 10e, ScaleTrack achieves the best performance across all metrics, with a substantial **AssocA** improvement of **+9.8** over MASA and **+18.2** over OVTrack. Notably, our model excels in **ClsA** (40.7), demonstrating its ability to track objects while maintaining strong semantic understanding. The consistent performance boost across all metrics validates the scalability of our contrastive learning framework.

Table 11: **Ablation on backbone version.**

| Backbone | # Videos | AssocA |
|----------|----------|--------|
| ResNet 50 | 1k | 55.1 |
| Swin-T | 1k | 54.7 |
| Swin-B | 1k | 55.8 |
| DINOv2-S | 1k | 57.2 |
| DINOv2-S | 12k | 60.3 |
| DINOv2-L | 12k | 61.3 |

## C.7 RESULTS ON ANIMALTRACK

AnimalTrack focuses on tracking non-rigid, freely moving objects, making it a challenging benchmark. As shown in Tab. 10f, our approach achieves competitive results, with a **TETA** of 57.2 and **AssocA** of 50.2. While MASA attains the highest **AssocA** (51.3), our method performs similarly while excelling in **ClsA** (54.8). These results confirm the effectiveness of our model in tracking highly deformable objects under natural conditions.

## C.8 RESULTS ON IMAGENETVID

ImageNetVID is a video extension of the ImageNet dataset, featuring high-quality object tracking annotations. As seen in Tab. 10g, ScaleTrack achieves the best **AssocA** (60.1), surpassing MASA by **+1.6** and OVTrack by **+16.7**. Our approach also achieves competitive **TETA** (55.4) and **ClsA** (37.2), confirming its strong association capability in densely labeled videos. These results further validate our method's ability to generalize across various tracking settings.

## C.9 SUMMARY

Across all benchmarks, ScaleTrack consistently achieves state-of-the-art results, particularly in **AssocA**, which is crucial for robust tracking. The improvements over prior methods demonstrate the effectiveness of our *Chain Contrastive Learning* approach in capturing long-term temporal dependencies while maintaining strong local discriminability. Our method successfully generalizes to diverse datasets, making it a highly scalable solution for real-world MOT applications.

# D MORE ABLATIONS

We here provide additional ablation studies on different backbones, the use of intermediate supervision and the depth of the transformer, the impact of pseudo-negative proposals, the effect of multi-scale features, efficiency comparisons, classic MOT17 results, SAM2-based baselines, a frame-sampling ablation, and a unified (detector–tracking head) variant.

When marked with †, we evaluate trackers with the *identical public detections* used by the compared baseline; only the association module differs. This isolates association quality and follows standard practice in MOT. Unless noted, inference uses a fixed input resolution of $518 \times 518$. FPS was measured on a single RTX 4090.

**Different backbones.** Tab. 11 reports ScaleTrack's performance with different backbones. Under the 1K training schedule, DINOv2-S achieves the best results, leveraging its robust self-supervised features. Scaling the backbone from DINOv2-S to DINOv2-L with a 12K video training schedule further improves performance, reaching state-of-the-art AssocA on the TAO benchmark.

**Intermediate supervision and transformer depth.** We analyze the effects of intermediate supervision and the number of transformer layers in Tab. 12. Intermediate supervision consistently improves association accuracy. Increasing decoder depth enhances performance, but the improvement saturates at two decoder layers, with further increases adding computational overhead without significant accuracy gains. Increasing transformer layers beyond two slightly decreases performance, possibly due to

Table 12: **Ablation on intermediate supervision and transformer depth.** Intermediate supervision improves association accuracy (AssocA), particularly for fewer transformer layers.

| Intermediate Supervision | Decoder Layers | AssocA↑ | FPS↑ (504×504) |
|---|---|---|---|
| No | 1 | 55.9 | 43.5 |
| Yes | 2 | **57.3** | 42 |
| No | 2 | 55.9 | 42 |
| Yes | 4 | 56.1 | 32 |
| No | 4 | 55.8 | 32 |

Table 13: **Ablation of pseudo-negative proposals.** Incorporating pseudo-negative proposals improves feature discriminability, leading to a significant boost in AssocA.

| Method | AssocA ↑ |
|---|---|
| w/o Pseudo-Negative Proposals | 53.5 |
| w/ Pseudo-Negative Proposals | **57.3** |

Table 14: **Ablation study on backbone feature levels.** Using multi-scale features from DINOv2 significantly outperforms single-scale features on TAO.

| Feature Extraction | AssocA↑ |
|---|---|
| Last-level only | 55.9 |
| Multi-scale features | **57.3** |

overfitting on the limited (1k videos) ablation training set. We thus choose a two-layer decoder with intermediate supervision as the optimal balance between efficiency and accuracy.

**Pseudo-negative proposals.** We evaluate the impact of incorporating pseudo-negative proposals during training. To avoid leakage of positives into the negative pool, we discard any SAM/Detic proposal with $IoU \geq 0.5$ to a GT box in the same frame prior to use as a negative. Tab. 13 shows that removing pseudo-negatives results in a drop of 3.8% in AssocA. By leveraging additional hard negative examples from instance segmentation models, our approach learns more discriminative object embeddings, reducing ID switches and improving the overall association.

**Impact of multi-scale features.** We evaluate the effectiveness of using multi-scale backbone features versus single-scale features in Tab. 14. Leveraging multi-scale representations from the frozen DINOv2 backbone yields notably better association accuracy (AssocA), highlighting the advantage of richer spatial context across multiple scales. Using only single-scale features from the backbone substantially degrades tracking performance, underscoring the importance of incorporating multi-scale information for robust feature representation.

**Efficiency vs. accuracy under a shared measurement.** Table 15 compares Params/GFLOPs against accuracy with identical measurement (518×518). Both SCALETRACK variants are substantially more efficient than heavy baselines while improving association quality (Tab. 15).

Table 15: **Compute vs. accuracy (val).** Params/GFLOPs per image; TAO/BDD report AssocA; SportsMOT reports HOTA.

| Method | Backbone | Params | GFLOPs | TAO AssocA ↑ | BDD AssocA ↑ | SportsMOT HOTA ↑ |
|---|---|---|---|---|---|---|
| SLAck-L (in-domain) | Swin-L | 260M | 290 | 41.8 | — | — |
| GLEE-Pro (in-domain) | ViT-L | 400M | 500 | 46.2 | — | — |
| MASA (Detic)† | Swin-B | 179M | 174 | 44.1 | 52.9 | 73.6 |
| **ScaleTrack-R50**† | R50 | **37M** | **33** | 50.2 | 54.0 | 79.4 |
| **ScaleTrack-S**† | DINOv2-S | **30M** | **38** | **51.9** | **56.0** | **80.6** |

*Note.* SCALETRACK-R50 outperforms MASA with ∼20% of the parameters and ∼19% of the GFLOPs; SCALETRACK-S improves further at similar compute (Tab. 15).

**Classic pedestrian MOT: MOT17 (val).** Despite being a generalist (zero-shot) model, SCALE-TRACK achieves strong HOTA/IDF1 on MOT17, indicating robust identity features on a domain-specific benchmark (Tab. 16).

Table 16: **MOT17 validation.** Specialist = in-domain; Generalist = zero-shot. "—" unreported.

| Method | HOTA ↑ | IDF1 ↑ | MOTA ↑ |
|---|---|---|---|
| *Specialist (in-domain)* | | | |
| GTR | 63.0 | 75.9 | 71.3 |
| ByteTrack | — | 79.7 | 76.7 |
| OC-SORT | 68.0 | 79.3 | 77.9 |
| MixSort-Byte | 69.4 | 81.1 | 79.9 |
| MixSort-OC | 69.2 | 80.6 | 78.9 |
| *Generalist (zero-shot)* | | | |
| Grounded-SAM2 | 47.5 | 54.1 | 43.0 |
| MASA[†] | 63.5 | 74.0 | 73.6 |
| **ScaleTrack (ours)**[†] | **70.0** | **78.4** | **77.3** |

**SAM2-based MOT baselines.** Across BDD100K and SportsMOT, SCALETRACK surpasses strong SAM2-based trackers, supporting that CCL with hard local negatives learns discriminative identities beyond per-object VOS (Tabs. 17 and 18).

Table 17: **BDD100K MOT (val).**

| Method | TETA ↑ | IDF1 ↑ | mHOTA ↑ |
|---|---|---|---|
| MeMOTR (in-domain) | 53.6 | 69.2 | 40.4 |
| SAM2-MOT (cls-8) | — | 70.8 | — |
| MASA[†] | 54.4 | 71.3 | 46.2 |
| **ScaleTrack (ours)**[†] | **56.3** | **73.3** | **46.6** |

Table 18: **SportsMOT (val).**

| Method | HOTA ↑ | IDF1 ↑ | MOTA ↑ |
|---|---|---|---|
| ByteTrack (in-domain) | 69.0 | 77.9 | 97.5 |
| Grounded-SAM2 | 66.1 | 70.2 | 91.4 |
| MASA[†] | 73.6 | 71.2 | 97.0 |
| **ScaleTrack (ours)**[†] | **80.6** | **85.3** | **96.0** |

**Sampling strategy.** Uniform frame sampling yields higher association accuracy and mirrors deployment where frames arrive at a constant rate (Tab. 19).

**Unified variant: detector–tracking head integration.** To demonstrate compatibility with a unified pipeline, we attach our association module as a tracking head on a frozen Detic backbone (Swin-B) and train end-to-end on TAO (val). The unified variant improves *AssocA* by +6.1 over MASA under identical detections (Tab. 20).

Table 19: **Sampling ablation (AssocA ↑).**

| Strategy | AssocA |
|---|---|
| Random sampling | 55.1 |
| **Uniform sampling** | **57.5** |

Table 20: **TAO (val): unified variant vs. MASA (Detic).**

| Method | TETA ↑ | LocA ↑ | AssocA ↑ | ClsA ↑ |
|---|---|---|---|---|
| MASA (Detic)[†] | 46.3 | 65.8 | 44.1 | 28.9 |
| **ScaleTrack-unified-Detic**[†] | **47.8** | 65.2 | **50.2** | 28.1 |

