# OpenReview forum: "Scaling Open-world Multiple Object Tracking"
_ICLR.cc/2026/Conference — ICLR 2026 Conference Withdrawn Submission_

### Official Review · Reviewer_wXyn · 2025-10-28

**Soundness:** 2
**Presentation:** 3
**Contribution:** 2
**Rating:** 4
**Confidence:** 3

**Summary:**

This paper proposes ScaleTrack, a transformer-based framework for open-world multiple object tracking (MOT) designed to scale efficiently on video datasets. The core contribution is the Chain Contrastive Learning (CCL), a temporal contrastive strategy that links positive samples across consecutive frames, maintaining both local discriminability and long-range temporal consistency. The model incorporates pseudo-negative sampling (from SAM/Detic proposals) and a multi-scale spatiotemporal attention module for robust association. Extensive experiments on TAO, BDD100K, SportsMOT, and OVT-B show consistent zero-shot performance gains over prior methods (e.g., MASA, GLEE, OVTrack), demonstrating strong scalability and generalization.

**Strengths:**

1. The proposed Chain Contrastive Learning effectively bridges local and global temporal contexts, addressing the scalability limitations of prior pairwise or global contrastive approaches.
2. The ScaleTrack is evaluated on multiple benchmarks, including ablation studies and scaling analyses (chain length, frozen backbone, etc.) to show its effectiveness.

**Weaknesses:**

- Besides open-world MOT datasets, the authors are encouraged to evaluate the proposed method on standard MOT benchmarks such as MOT17, MOT20, and DanceTrack to more comprehensively demonstrate its association capability and general applicability.
- It is recommended to directly compare ScaleTrack with methods that do not use Chain Contrastive Learning (CCL), such as MeMOTR and MOTIP, to quantitatively verify the effectiveness of the proposed CCL module.
- The authors could further transfer or plug the proposed CCL mechanism into other existing architectures to verify its universality and adaptability beyond the presented framework.
- An efficiency comparison (inference speed, FLOPs, or training cost) should be added to better illustrate the practical scalability and computational trade-offs of the proposed approach.

**Questions:**

refer to weaknesses.

---

### Official Review · Reviewer_Te2h · 2025-10-30

**Soundness:** 2
**Presentation:** 2
**Contribution:** 2
**Rating:** 2
**Confidence:** 5

**Summary:**

The paper proposed a novel approach called chain contrastive learning to capture local discriminative features and long-range temporal coherence features from large-scale, sparsely annotated video data for open-world multople object tracking. Comprehensive experiments are conducted to effectiveness of the proposed method and its superiority to other state-of-the-art algorithms.

**Strengths:**

1. The problem of learning effective feature from sparsely annotated video data for open-world multiple object tracking is of importance in real-world application.
2. Experiment part is solid and demonstrates the effectiveness of the proposed method.

**Weaknesses:**

1. The writing and organization of the paper is moderate which makes it not easy to follow and understand the motivation of the proposed idea and method.
2. What are the challenges of "leverage large-scale, sparsely annotated video data".
3. Line 40, the author states that "...appearance association module as the most viable path toward generalizable MOT". How to come to this
conclusion, any evidence to support it?
4. Line 76, the author states that "As the data size increases...causing the learning to plateau." How to demonstrate this problem?

**Questions:**

see weaknesses

---

### Official Review · Reviewer_duKa · 2025-10-31

**Soundness:** 2
**Presentation:** 3
**Contribution:** 1
**Rating:** 2
**Confidence:** 5

**Summary:**

The paper proposes Chain Contrastive Learning to maintain local discriminability while capturing long-range temporal coherence. The performance seems good on many datasets.

**Strengths:**

The performance is good and the experiment is extensive.

**Weaknesses:**

- The paper claims to “scale up” OW-MOT, but it is unclear in which aspect this scaling occurs. The proposed approach does not appear to expand the category space, task complexity, or range of scenarios typically associated with open-world settings. Instead, the work primarily modifies the contrastive learning formulation. Therefore, the claim of scaling up OW-MOT seems overstated and should be clarified or reframed to accurately reflect the scope of the contribution.

- The paper mentions experiments on partially labeled videos, but these scenarios are not clearly demonstrated or discussed. It remains unclear what proportion of the training data is labeled versus unlabeled, and how the proposed method effectively leverages the unlabeled portions. If it is about pseudo-negative samples, this can be done simply by selecting proposals other than positive by thresholding, similar to QDTrack [1]. The only data scaling comparison in Figure 3 was not conducted with other SOTA works.

- The proposed chain formulation, which accumulates pairwise similarities across frames via summation, appears conceptually similar to existing holistic temporal association strategies. For instance, QDTrack already encourages transitive consistency across multiple positives over entire video sequences. The novelty and benefit of the proposed method over such established approaches are therefore not well justified.

- Initially, I had the feeling that the chain could be extended to infinite length via online processing; however, taking a look at Table 5. It seems that the chaining methodology only supports a handful number of samples. So it does not make a significant difference to batch processing in Figure 1a.

[1] Fischer, et al. Qdtrack: Quasi-dense similarity learning for appearance-only multiple object tracking.TPAMI 2023.

**Questions:**

See weaknesses.

---

### Official Review · Reviewer_7EJo · 2025-11-01

**Soundness:** 3
**Presentation:** 3
**Contribution:** 2
**Rating:** 4
**Confidence:** 4

**Summary:**

This work proposes a new method for open-world multiple object tracking. It proposes a new contrastive strategy for better contrastive learning. A transformer-based model is built utilizing this strategy. Experiments on multiple public benchmarks demonstrate the effectiveness of the proposed method.

**Strengths:**

1. The idea of selecting samples from constrained frames for contrastive learning makes sense.
2. The implementation of the chain constrastive learning strategy is reasonable.
3. The proposed method is effective on multiple public benchmarks.

**Weaknesses:**

The main concern is limited methodology contribution. The main contribution is the chain contrastive learning strategy, which is a straightforward modification of existing strategies. The corresponding ScaleTrack generally adopts existing modules. Hence the contribution on the methodology part is not enough from my view.

**Questions:**

I am curious about the universality of the proposed contrastive learning strategy. Could it be applied to other trackers, besides the ScaleTrack? It would be great if related experiments are included.

---

### Note · Authors · 2025-11-14

I have read and agree with the venue's withdrawal policy on behalf of myself and my co-authors.